# Modeling the Machine Learning Multiverse

**Samuel J. Bell**[1]    **Onno P. Kampman**[2]    **Jesse Dodge**[3]    **Neil D. Lawrence**[1]
[1]Computer Laboratory, University of Cambridge
[2]Department of Psychology, University of Cambridge
[3]Allen Institute for AI
{sjb326,opk20,ndl21}@cam.ac.uk   jessed@allenai.org

## Abstract

Amid mounting concern about the reliability and credibility of machine learning research, we present a principled framework for making robust and generalizable claims: the *multiverse analysis*. Our framework builds upon the multiverse analysis [1] introduced in response to psychology's own reproducibility crisis. To efficiently explore high-dimensional and often continuous ML search spaces, we *model* the multiverse with a Gaussian Process surrogate and apply Bayesian experimental design. Our framework is designed to facilitate drawing robust scientific conclusions about model performance, and thus our approach focuses on exploration rather than conventional optimization. In the first of two case studies, we investigate disputed claims about the relative merit of adaptive optimizers. Second, we synthesize conflicting research on the effect of learning rate on the large batch training generalization gap. For the machine learning community, a multiverse analysis is a simple and effective technique for identifying robust claims, for increasing transparency, and a step toward improved reproducibility.

## 1   Introduction

Machine learning research faces mounting concern about the reliability of our results and the credibility of our claims [2–20]. The field of psychology has faced a similar crisis [21], and confrontation with its shortcomings has sparked practical innovations [1, 22–25]. One such innovation is of direct relevance to the machine learning community: the *multiverse analysis* [1].

Throughout any investigation, scientists make decisions about how to perform their work. In psychology, as in other disciplines, there are a plethora of different ways to conduct and analyze experiments. From just a handful of choices, we reach such a large decision tree that researchers can repeatedly try different paths until they chance upon a positive result [26]. Even without conscious manipulation, these decision points pose a fundamental problem: what if the psychologist had chosen an alternate—and perfectly reasonable—route through this garden of forking paths [27]? Would their results and conclusions still stand?

In their replication of a contentious study [28] on the effect of menstrual cycle and relationship status on women's political preferences, Steegen et al. [1] identify both decisions taken and reasonable alternatives.[1] They name the Cartesian product of alternatives a *multiverse*, a set of parallel universes each containing a slightly different study. Introducing the *multiverse analysis*, Steegen et al. re-run the study as if inside each universe, finding that the alleged effect of relationship status and fertility is highly sensitive to different choices, in particular the definition of "single". Multiverse analyses have subsequently been used to evaluate the robustness of claims across a variety of psychological and neuroscientific settings (e.g. [29–33]).

---

[1]E.g. outlier selection; variable discretization; and how to estimate menstrual onset.

36th Conference on Neural Information Processing Systems (NeurIPS 2022).

In a machine learning context, we present the multiverse analysis as a principled framework for analyzing robustness and generality. Consider an example of some modification to a model, say batch normalization [34]. To verify batch norm's efficacy, one needs a test bed including model architecture, optimization method, dataset, evaluation metric, and so on. Regardless of these specific choices, we would like that batch norm be effective in general. With a multiverse analysis, we can systematically explore the effect of each choice, and understand the circumstances in which a claim holds true.

Our primary contribution is to introduce the multiverse analysis to ML, which we use to draw more robust conclusions about model performance. To efficiently explore the high-dimensional and often continuous ML search space, we model the multiverse with a Gaussian Process (GP) surrogate and use Bayesian experimental design (§ 2). We present motivating evidence that choosing exploration over optimization—the essence of a multiverse analysis—is essential when we seek proper understanding of our claims and their generality (§ 3). In the first of two case studies, we use a multiverse analysis over hyperparameters to replicate an experiment on adaptive optimizers [35], finding that claims on the relative merits of optimizers are highly sensitive to learning rate (§ 4). Second, we perform a more extensive multiverse analysis to synthesize divergent research on the large batch "generalization gap" [36], finding an interaction effect of batch size and learning rate (§ 5). We conclude by discussing the limitations of our approach and directions for future work (§ 6).

## 2 Efficient multiverse exploration

Our approach to the multiverse analysis packages ML researcher decisions into a simple framework, requiring only a choice of **search space** $\mathcal{X}$ and an **evaluation function** $\ell$, which together define our multiverse. This search space defines the set of *reasonable* choices. For example, we might consider only a few directly relevant hyperparameters (§ 4); include purportedly irrelevant choices such as dataset (§ 5); or conduct even more expansive analyses (§ 6). The evaluation function should pertain to the hypothesis being tested. Here, we define it as model test accuracy (§ 5) or difference in test accuracy (§ 4).

A barrier to analyzing any ML multiverse is the size of the search space and the presence of continuous dimensions. This makes exhaustive search intractable. To overcome this, we use a GP surrogate—modeling $\ell$ as a function of $\mathcal{X}$—as a stand-in for the multiverse. Using the surrogate, we iteratively explore the space using Bayesian experimental design [37]:

1. Sample an initial design, $X_0 \sim \mathcal{X}$, and evaluate $\ell$ at each point, $Y_0 = \ell(X_0)$;

2. Fit a GP model $f$ to the sampled points $X_0$ and corresponding results $Y_0$;

3. Use an acquisition function $a$ on $f$ to sample and evaluate a new batch $(X_i, Y_i)$;

4. Repeat steps 2–3 until an appropriate stopping criterion.

In step 1, our initial design is drawn from a Sobol sequence, a low discrepancy sequence that achieves improved coverage over uniform random sampling in higher dimensions [38]. The search space and the evaluation function are specific to the analysis at hand and described later.

In step 2, we model the output of our evaluation function as a noisy function of the inputs,

$$\mathbf{y}_i = f(\mathbf{x}_i) + \epsilon_i, \quad \epsilon_i \sim \mathcal{N}(0, \Sigma). \tag{1}$$

Placing a GP prior over $f$ with zero mean and positive definite kernel function $k$,

$$f \sim \mathrm{GP}(0, k), \tag{2}$$

we obtain the posterior mean $\mu$ and variance $\sigma^2$ [39, chap. 2]:

$$\mathbf{y}_{i+1} \sim \mathcal{N}(\mu(\mathbf{x}_{i+1}), \sigma^2(\mathbf{x}_{i+1})), \tag{3}$$

$$\mu(\mathbf{x}_{i+1}) = \mathbf{k}^T K^{-1} Y_i, \tag{4}$$

$$\sigma^2(\mathbf{x}_{i+1}) = k(\mathbf{x}_{i+1}, \mathbf{x}_{i+1}) - \mathbf{k}^T (K + \Sigma)^{-1} \mathbf{k}, \tag{5}$$

where $K$ is the kernel matrix and $\mathbf{k} = [k(\mathbf{x}_{i+1}, \mathbf{x}_0), \ldots, k(\mathbf{x}_{i+1}, \mathbf{x}_i)]$. We use a Matérn 5/2 kernel with automatic relevance determination [40].

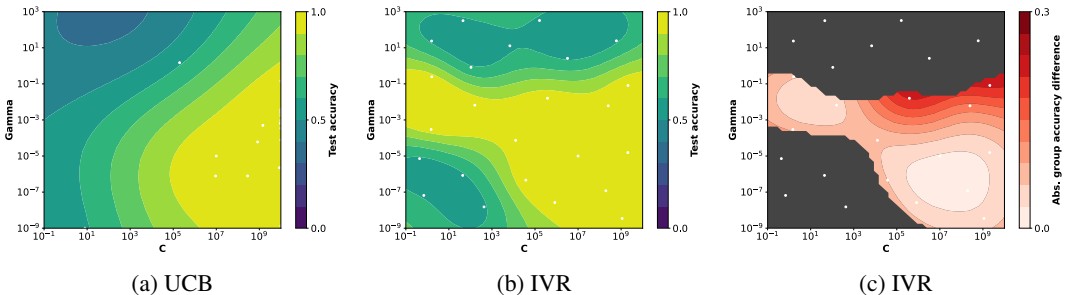

| (a) UCB | (b) IVR | (c) IVR |

Figure 1: Contour plot of GP-predicted mean test accuracy over search space of $C$ and $\gamma$ (Gamma) as explored by **(a)** UCB and **(b)** IVR acquisition functions. **(c)** Secondary objectives, e.g. minimizing group-level outcome differences, may vary along the IVR-revealed plateau.

In step 3, we use integrated variance reduction (IVR) as our acquisition function [41] (motivated in § 3). IVR calculates the change in *total* variance if we were to acquire a candidate point $\mathbf{x}_{i+1}$

$$a(\mathbf{x}_{i+1}; X_i, Y_i) = \int_{\mathcal{X}} \sigma^2(\mathbf{p}; X_{i+1}, Y_{i+1}) - \sigma^2(\mathbf{p}; X_i, Y_i)\, d\mathbf{p}\,, \qquad (6)$$

where $X_i, Y_i$ excludes the candidate point and $X_{i+1}, Y_{i+1}$ includes it.

Using a Monte Carlo approximation[2] for the intractable integral, the most informative point $\mathbf{x}*$ to sample next is

$$\mathbf{x}* = \underset{\mathbf{x}_{i+1}}{\arg\min}\, a(\mathbf{x}_{i+1}; X_i, Y_i)\,. \qquad (7)$$

A key difference between standard hyperparameter optimization and our framework is that when optimizing, all models except the best performing are discarded, whereas we use all available information to draw more robust conclusions. To do this, we continue to use our surrogate model. First, we visualize the multiverse using the surrogate's posterior predictive mean, and visualize the extent to which we have explored through the surrogate's posterior variance. Second, we test for interaction effects by comparing a GP with a shared kernel $M_{\text{shared}}$ against an additive kernel $M_{\text{additive}}$ (see fig. S1). We compare their fit using the Bayes factor $K$,

$$K = \frac{P(X, Y | M_{\text{additive}})}{P(X, Y | M_{\text{shared}})}\,, \qquad (8)$$

where $P(X, Y | M)$ is the marginal likelihood of the observations given a model. $K > 1$ indicates that $M_{\text{additive}}$ better explains the data than $M_{\text{shared}}$, indicating the absence of an interaction. This also provides an appropriate termination condition for step 4: we continue to sample until we reach a conclusive Bayes factor. Finally, we perform a Monte Carlo sensitivity analysis [42, 43] to assess how much a change in one of the parameters would affect the model outputs.

For GP modeling we use GPy [44] with EmuKit [45] for experimental design and sensitivity analysis. We use TorchVision's [46] off-the-shelf deep learning model architectures.

## 3  Motivating example: SVM hyperparameters

Here we motivate the multiverse analysis and our focus on exploration with an analysis of hyperparameter optimization. To both aid replication and assess the effect of hyperparameter tuning on final results, there are calls for increased transparency around search space and tuning method (e.g. [47]). Going one step further, we argue that *premature* optimization—even if transparently reported—hinders our understanding. We illustrate this with a simple example, tuning two hyperparameters of an SVM classifier to show that optimizing leads to a distorted view of hyperparameter

---

[2]In particularly high-dimensional search spaces, it may be more appropriate to use a quasi-Monte Carlo method to estimate the variance reduction over a sample of points with improved coverage of the space.

space. We compare optimization and exploration using the upper confidence bound (UCB) [48] and IVR acquisition functions.

In this multiverse analysis, we define the **search space** as the SVM's regularization coefficient $C$ and the lengthscale $\gamma$ of its RBF kernel. Our **evaluation function** is the test accuracy of the SVM on the Wisconsin Breast Cancer Dataset [49] of samples of suspected cancer to be classified as benign or malignant. Given the same initial sample, we evaluate 23 further configurations for each acquisition function.

If we optimize, we would conclude from fig. 1a that there is a single region of strong performance. Conversely, exploring with IVR (fig. 1b) shows that this region is in fact a plateau, and we are free to use any value of $C$ as long as we scale $\gamma$ accordingly. While best test accuracy is similar in both cases, only by exploring do we learn about the full space of our options. This knowledge is of vital importance if we properly account for additional real-world objectives, such as minimizing disparity in outcomes across different social groups. We test this idea by assigning a synthetic majority/minority group label, $g \sim \mathrm{Bern}(0.4)$, to each datapoint, and show in fig. 1c how group-level outcomes can vary along the plateau revealed by IVR. Prematurely optimizing could easily result in selecting a model that introduces group disparity.

There is, of course, an appropriate moment for optimization. As one moves along the spectrum from research to deployment, so too should one move from exploration to exploitation. Our aim here is not to critique optimization *per se*, but to highlight that exploration is also of paramount importance. When conducting scientific research in particular, we argue it is more appropriate to learn and understand as much possible, rather than eke out another marginal improvement. Here we use a multiverse analysis as a framework for systematic exploration.

## 4 Case study 1: When are adaptive optimizers helpful?

Adam [50] is a popular adaptive optimizer for training deep neural networks. Questioning this practice, Wilson et al. [35] suggest adaptive optimizers offer limited advantages over vanilla stochastic gradient descent (SGD) [51]. They present experiments showing that SGD with momentum [52] outperforms all other optimizers across image recognition on CIFAR-10 [53], language modeling, and constituency parsing.

Under certain hyperparameter conditions, however, adaptive optimizers can be considered equivalent to SGD with momentum, highlighting the crucial role of hyperparameter tuning [54]. In replications of Wilson et al.'s experiment with VGG [55] on CIFAR-10, additionally tuning Adam's $\epsilon$ hyperparameter—introduced solely for numerical stability and typically ignored—eliminates SGD's advantage [6, 54].

Both replications [6, 54] can be considered proof by existence. In order to refute Wilson et al.'s claim, it suffices to find any point in hyperparameter space where Adam beats SGD with momentum.

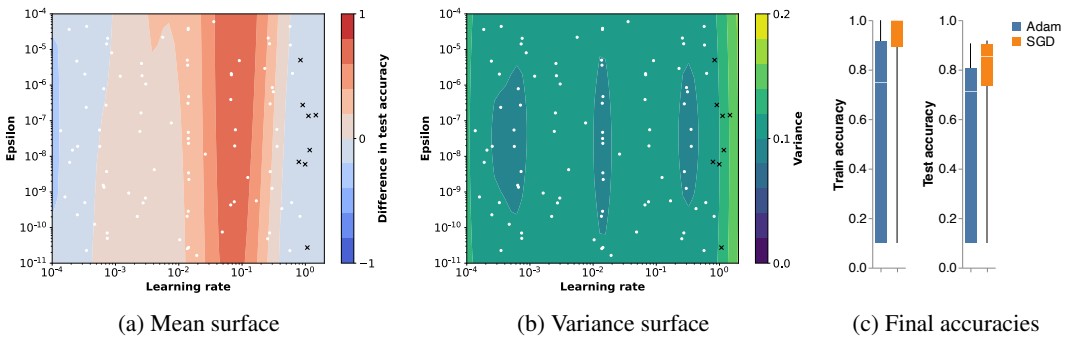

(a) Mean surface          (b) Variance surface          (c) Final accuracies

Figure 2: Contour plot of GP-predicted **(a)** mean difference in test accuracy (SGD - Adam) and **(b)** variance over the search space of learning rate and $\epsilon$. Red regions indicate SGD with momentum outperforms Adam. White points are successful trials; black crosses failed. **(c)** Final train and test accuracies. Whiskers extend to min and max. Note SGD train accuracy has median, UQ and max $1.0$.

However, this approach tells us little about relative optimizer performance *in general*. In this short case study, we perform a multiverse analysis of Wilson et al.'s experiment with VGG on CIFAR-10. We explore a reasonable and relevant search space and analyze how different hyperparameter choices lead to different optimizer recommendations.

### 4.1 Multiverse definition

Because the claim under scrutiny compares optimizers, we define the **evaluation function** as the difference between model test accuracy achieved by Adam and SGD with momentum. Let $\mathbf{g}_t = \nabla f(\theta_{t-1})$ be the minibatch-estimated gradient of the loss function $f$ w.r.t. the model parameters $\theta_{t-1}$. For SGD with momentum, we take a step of size $\alpha$ in the direction of a decaying sum of recent gradients,

$$\theta_t = \theta_{t-1} - \alpha \mathbf{d}_t \,, \quad \mathbf{d}_t = \mu \mathbf{d}_{t-1} + \mathbf{g}_t \,,$$

where momentum parameter $\mu$ controls decay. In contrast, Adam steps along the gradient normalized by an unbiased estimate of its first and second moments $\hat{\mathbf{m}}_t$ and $\hat{\mathbf{v}}_t$:

$$\mathbf{d}_t = \frac{\hat{\mathbf{m}}_t}{\sqrt{\hat{\mathbf{v}}_t} + \epsilon} \,,$$

$$\hat{\mathbf{m}}_t = \frac{\mathbf{m}_t}{1 - \beta_1^t} \,, \quad \mathbf{m}_t = \beta_1 \mathbf{m}_{t-1} + (1 - \beta_1)\mathbf{g}_t \,, \quad \mathbf{m}_0 = 0 \,,$$

$$\hat{\mathbf{v}}_t = \frac{\mathbf{v}_t}{1 - \beta_2^t} \,, \quad \mathbf{v}_t = \beta_2 \mathbf{v}_{t-1} + (1 - \beta_2)\mathbf{g}_t^2 \,, \quad \mathbf{v}_0 = 0 \,.$$

Like Wilson et al., we use the default of $\mu = 0.9$ and use the Polyak implementation. For Adam, we also use default parameters $\beta_1 = 0.9$ and $\beta_2 = 0.999$. The model is VGG-16 with batch normalization [34] and dropout [56], trained for 300 epochs on CIFAR-10.

We set our **search space** to learning rate $\alpha \in [10^{-4}, 10^0]$ by $\epsilon \in [10^{-11}, 10^{-4}]$. $\epsilon$ is only applied to Adam. We evaluate 3 batches of 32 points.

### 4.2 Results

The contour plot in fig. 2a shows a large region ($10^{-3} \leq \alpha \leq 10^{\frac{1}{2}}$) in which SGD outperforms Adam. However, we also identify regions (approx. $\alpha < 10^{-3}$ or $\alpha > 10^{\frac{1}{2}}$) where the opposite is true, though we note the higher uncertainty (fig. 2b) in the high learning rate region. See fig. S2 for raw results.

The change in relative performance is described almost entirely by learning rate. The sensitivity analysis (fig. 3) reveals main effects (i.e. sensitivity to each variable in isolation) for learning rate and $\epsilon$ of $0.990 \pm 0.014$ and $-0.004 \pm 0.0002$, and similarly total effects (i.e. sensitivity to each variable including its interaction with others) of $0.996 \pm 0.014$ and $0.008 \pm 0.031$. Testing for interaction effects via Bayes factor (see eq. (8)), we find $K = 2635$, thus rejecting the existence of an interaction between learning rate and $\epsilon$.

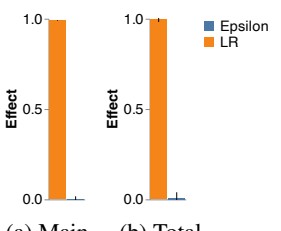

(a) Main  (b) Total

Figure 3: **(a)** Main and **(b)** total effects of learning rate and $\epsilon$. LR drives almost all output variance. Bars are STD.

In summary, we find that SGD with momentum often, but not always, outperforms Adam when training VGG on CIFAR-10, and that the conclusion we draw is highly sensitive to selected learning rate. Unlike others [6, 54], we do not find a significant effect of $\epsilon$, instead finding that learning rate determines which optimizer performs best. In practice, our results suggest optimizer choice may make little difference to final test error, as long as an appropriate learning rate is used. If budget for hyperparameter search is limited, $\epsilon$ is unlikely to be the most efficient hyperparameter to include in a search.[3] Using a multiverse analysis, we have systematically explored how decisions about learning rate and epsilon impact conclusions about the optimal optimizer. In doing so, we have shown that reported results may vary according to researcher choices.

---

[3]That said, our results do not preclude the possibility that with a much more extensive and optimization-based search, the resulting estimate of the search space might reveal some (modest) room for improvement from tuning $\epsilon$. However, as indicated by the sensitivity analysis in fig. 3, any effect is likely to be dwarfed by properly tuning learning rate, and is unlikely to materially alter the best choice of optimizer.

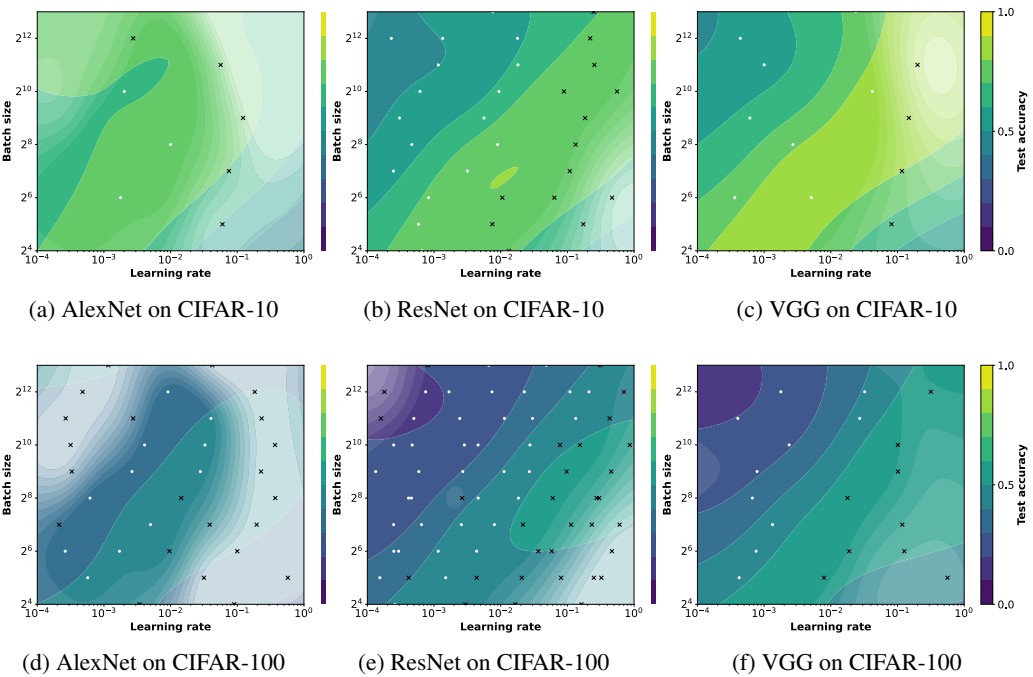

| | | |
|---|---|---|
| (a) AlexNet on CIFAR-10 | (b) ResNet on CIFAR-10 | (c) VGG on CIFAR-10 |
| (d) AlexNet on CIFAR-100 | (e) ResNet on CIFAR-100 | (f) VGG on CIFAR-100 |

Figure 4: Contour plot of GP-predicted mean test accuracy over the search space of learning rate, batch size, dataset and model. White points are trials with training accuracy $\geq 0.99$; black crosses were excluded. Overlayed translucent regions indicate high training error. For Tiny ImageNet see fig. S4; for variance see fig. S5. The discrepancy between contours and data points in **(a)** is due to the coregionalized model sharing information across functions.

## 5 Case study 2: Is there a large-batch generalization gap?

Larger batch sizes are a prerequisite for distributed neural network training. However, many researchers have reported a "generalization gap", where generalization performance declines as batch size increases [36, 57, 58], a phenomenon potentially [59-62] but not conclusively [63] linked to the sharpness of the resulting minima.

Subsequent work has sought ways to mitigate the generalization gap by, in broad strokes, either scaling up the batch size throughout training [64, 65, pp. 262-265], or by scaling learning rate proportional to batch size [66-69]. To make sense of these results, one must cut through myriad researcher choices about datasets; model architectures; termination criteria [70]; linear [68] or sublinear (e.g. square root) [66] scaling rules; high initial learning rate [66]; learning rate decay or warmup [68]; layer-specific learning rates [69]; batch norm variants [66]; regularization techniques e.g. label smoothing [70]; and so on. In our second case study, we use a multiverse analysis to synthesize existing research on the relationship between batch size, learning rate and generalization error.

### 5.1 Multiverse definition

Our **evaluation function** is the test accuracy of a model trained according to the sampled configuration. The **search space** includes learning rate $\alpha \in [10^{-4}, 10^{-\frac{1}{2}}]$, batch size $\in \{2^4, \ldots, 2^{13}\}$, model $\in \{$AlexNet [71], VGG [55], ResNet [72]$\}$, and dataset $\in \{$CIFAR-10, CIFAR-100 [53], Tiny ImageNet[73]$\}$. Specifically, we use VGG-16 with batch norm and ResNet-18. Tiny ImageNet was selected as a substitute for ImageNet [74] to limit compute expenditure.

For categorical parameters, we treat each pair of model and dataset as its own function with an intrinsic coregionalization model [75, 76]. Given base kernel $k$, our multi-output kernel matrix is:

$$B_m = \mathbf{w}_m\mathbf{w}_m^\top + \text{diag}(\kappa_m),$$
$$B_d = \mathbf{w}_d\mathbf{w}_d^\top + \text{diag}(\kappa_d),$$
$$K(X,X) = B_m \otimes B_d \otimes k(X,X),$$

where $B_d$ and $B_m$ represent dataset and model respectively. In interpreting these parameters, the outer product of $\mathbf{w}$ defines how related each output is, whereas $\kappa$ allows them to vary independently.

We evaluate 6 batches of 32 points. Before analysis, we discard all model runs with training accuracy $< 0.99$ and re-fit the model. In addition to previous methods we also analyze the coregionalization parameters to investigate how much dataset and model impact our results.

## 5.2 Results

Of 224 trials, 9 failed to converge and a further 115 did not reach training accuracy $\geq 0.99$. A preliminary analysis of the raw results (see fig. S3) shows test accuracy is barely positively correlated with learning rate ($\rho = 0.08$) and slightly negatively correlated with batch size ($\rho = -0.2$). In fig. 4, however, a consistent plateau emerges across all models on both CIFAR-10 and CIFAR-100: test accuracy is maintained as long as batch size scales up with learning rate. On both sides of this plateau, if either learning rate or batch size are too large, generalization error will increase.

We were unable to successfully train any model on Tiny ImageNet. Of the 42 models we trained on Tiny ImageNet, 22 reached the 0.99 training accuracy threshold, but none of these obtained a test accuracy higher than 0.01, indicating significant overfitting. See fig. S4 for Tiny ImageNet contours.

Testing for interaction effects via Bayes factor (see eq. (8)), we find $K = 0.58$, suggesting that an additive kernel is not sufficient to explain the data and indicating an interaction effect. This implies that batch size alone does not explain generalization, but batch size and learning rate *together*. Our modeling approach gracefully handles the failures on Tiny ImageNet, such that even with Tiny ImageNet trials removed, the interaction effect remains present ($K = 0.34$).

The coregionalization kernels explain how model and dataset affect our results. In fig. 5b all model outputs are highly correlated, indicating that choice of model does not impact the relationship between batch size, learning rate, and generalization error. In contrast fig. 5a shows highly correlated outputs across CIFAR-10 and CIFAR-100, and moderate negative correlation with Tiny ImageNet. A strength of our chosen coregionalized model is that it flexibly captures the batch size and learning rate relationship in CIFAR-10 and 100, despite the failures on Tiny ImageNet. Our results on CIFAR-10 and CIFAR-100 indicate that the relationship may hold across datasets *where training was successful*, including across different dataset complexities, in contrast to previous work [57].

Finally, our sensitivity analysis in fig. 6 reveals next to no main effect for either learning rate ($0.03 \pm 0.06$) or batch size ($0.02 \pm 0.06$). However, the larger total effects for both learning rate ($0.19 \pm 0.11$) and batch size ($0.15 \pm 0.11$) again support the existence of an interaction effect. Both learning rate and batch

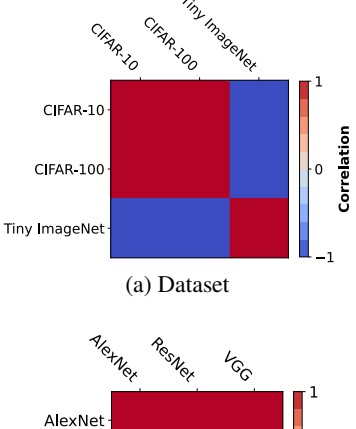

(a) Dataset

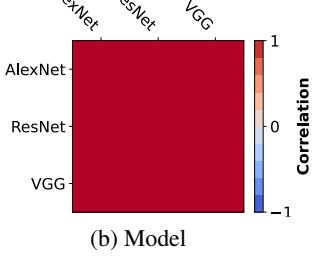

(b) Model

Figure 5: Multi-output GP function correlations. **(a)** Tiny ImageNet is moderately negatively correlated with CIFAR-10/100. **(b)** All three model outputs are highly correlated.

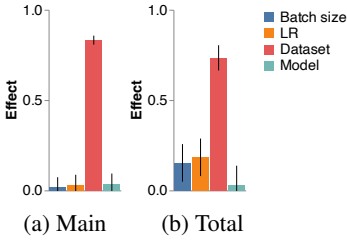

(a) Main     (b) Total

Figure 6: **(a)** Main and **(b)** total effects of all parameters. LR drives more output variance than batch size, but dataset is most important. Increased total effect for LR and batch size confirms interaction effect. Bars are STD.

size are dominated by the main effect of dataset ($0.83 \pm 0.03$), though this is likely due to Tiny ImageNet.

In conclusion, our framework reveals a complex interaction between learning rate, batch size, and generalization error that is consistent across all models and two out of three datasets. These results support scaling learning rate linearly with batch size [61, 67, 68], but do not support a threshold batch size (up to $2^{13}$) after which learning rate is no longer corrective [57, 70]. After accounting for learning rate, we find no consistent evidence of the proposed [36, 58] large-batch generalization gap. Here, a multiverse analysis has proven a useful tool to pull together disparate research on the generalization gap and to surface underlying trends.

# 6   Discussion

In the three applications of the multiverse analysis presented here, we have demonstrated that decisions taken by the researcher can have a significant effect on the final conclusions drawn. In our SVM hyperparameter example, we highlight the important role of systematic exploration and show that premature optimization may both limit our understanding and present only a partial truth to downstream practitioners. Our first case study on optimization demonstrated how varying the learning rate results in fundamentally different conclusions about whether to use SGD with momentum or Adam for optimization. Conversely, our second case study showed a simple relationship between batch size, learning rate and generalization error irrespective of model and dataset.

## 6.1   Choice of search space

Across these examples we have tried to illustrate differently complex analyses by way of differently sized search spaces. In case study 1 we inflated a rather small multiverse focusing only on directly relevant hyperparameters, though in case study 2 we built a larger multiverse including parameters (model and dataset) that vary freely between existing research. An expected critique of multiverse analyses is that choice of search space is itself a choice.[4] However, choosing dimensions and bounds for a search space is more principled than choosing specific points, and that declaring each dimension makes assumptions of relevance or irrelevance explicit. That said, it is always possible for subsequent work to critique the choice of search space and to add new dimensions. Indeed, our first case study *should* be expanded to include each of Wilson et al.'s experiments with different model architectures, datasets, deep learning frameworks and additional adaptive optimizer variants, though we reinforce that we chose a limited analysis to provide a simple case study. Our second case study also presents a number of interesting avenues for expansion in subsequent analyses, including additional datasets and model architectures, learning rate schedules and optimizers. Most interesting would be inclusion of termination criterion and evaluation metric, both previously highlighted as key drivers of divergent findings [70].

## 6.2   Compute cost

Our main contribution is to present the multiverse analysis framework and show how it can be used to draw robust conclusions about model performance, so we allocated our compute budget to showcase illustrative examples. Our framework is equally appropriate for research pushing state-of-the-art with a larger budget, though such experiments aren't necessary to demonstrate our framework's value. In this project, we used 1138 hours of GPU time, at a rough cost of \$775 with $140\,\mathrm{kg}$ of $CO_2$.[5] We expect costs to scale with multiverse size, requiring a pause for consideration in light of recent critiques of the environmental impacts of ML [77–79]. In response, we first note that at least some of the required compute is already taking place as part of existing trial-and-error workflows. Second, through the introduction of a surrogate model for the multiverse, one can substantially reduce the amount of exploration required. Finally, we suggest that *all* resources consumed in producing non-robust results, and those of subsequent work that builds upon them, are wasted by definition.

---

[4]The same applies to GP surrogate setup, e.g. kernel choice, though we suggest our settings used in this work will be a suitable default for most purposes. See figs. S6 to S10 for a comparison of different kernels.

[5]Calculated using `https://mlco2.github.io/impact` [77] assuming A100 GPUs on the University of Cambridge HPC cluster with carbon efficiency $0.307\,\mathrm{kgCO_2/kWh}$.

### 6.3 Future work

It is common practice when reporting model performance to train a number of runs with different random seeds and report the mean and standard deviation. Number of runs varies according to researcher budget and time. Appropriate sample size to enable robust inference in the face of noise is rarely considered. Our approach to the multiverse analysis presents an elegant potential solution to this issue. In our examples thus far we train only a single model for each point and assume homoscedasticity for simplicity. However, research in experimental design suggests that accounting for heteroscedasticity by separately modeling the variance in the surrogate [80, 81] could allow for a principled trade-off between conducting another run and a sampling a new configuration [82].

While we introduce the multiverse analysis to ML, it has previously been applied to a handful of studies in human computer interaction [83]. In an exciting area for development, the authors also develop interactive visualizations to help the reader explore the effects each choice, up to and including re-rendering written conclusions in an online version of the paper. Given that the largest multiverse in our work is 4-dimensional, our faceted contour plots are sufficient to communicate key trends. However, in more expansive multiverses novel visualization techniques will become essential.

Recent experiments with pre-registration in ML (e.g. [84]) promise to enforce a distinction between exploratory and confirmatory research [23, 85]. Committing to an experimentation and analysis plan in advance can be a helpful foil for questionable practices such as tweaking the parameters until one reaches a positive result. However, this commitment is limited to a single analysis—one particular instantiation of a decision set—while the other possible analyses, the remainder of the multiverse, remain unexplored. We see the multiverse analysis as complementary to pre-registration, in that pre-registering a multiverse analysis both codifies auxiliary hypotheses and allows room for exploration.

## 7 Conclusion

For continued progress in ML, we depend on reproducible results and conclusions that generalize to new settings. We have introduced the multiverse analysis as a principled framework to explore the impact of researcher decisions and facilitate the drawing of more robust conclusions. Our first case study unifies conflicting work on the benefit of adaptive optimizers and reinforces that optimizer merit is driven primarily by learning rate. In our second case study we identified a complex interaction between batch size, learning rate and generalization error, and we dispute the existence of a generalization gap. We have also presented evidence supporting the practice of scaling learning rate linearly with batch size.

By using a multiverse analysis, both researchers and practitioners gain more robust claims, better understanding of how decisions impact results, and helpful insight into the generality and reproducibility of conclusions.

## Acknowledgments and Disclosure of Funding

SB is supported by the Biotechnology and Biological Sciences Research Council [grant number BBSRC BB/M011194/1]. NL is supported by a Senior Turing AI Fellowship funded by the UK government's Office for AI, through UK Research and Innovation (grant reference EP/V030302/1), and delivered by the Alan Turing Institute. NL's chair is endowed by DeepMind.

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
