# Appendix

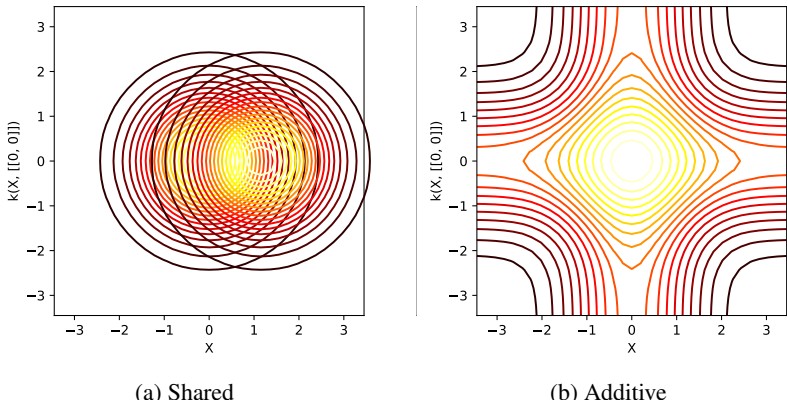

(a) Shared

(b) Additive

Figure S1: Illustration of (a) shared and (b) additive kernel. We use the additive kernel to model the absence of interaction effect, as it computes similarity along either (rather then both) dimension.

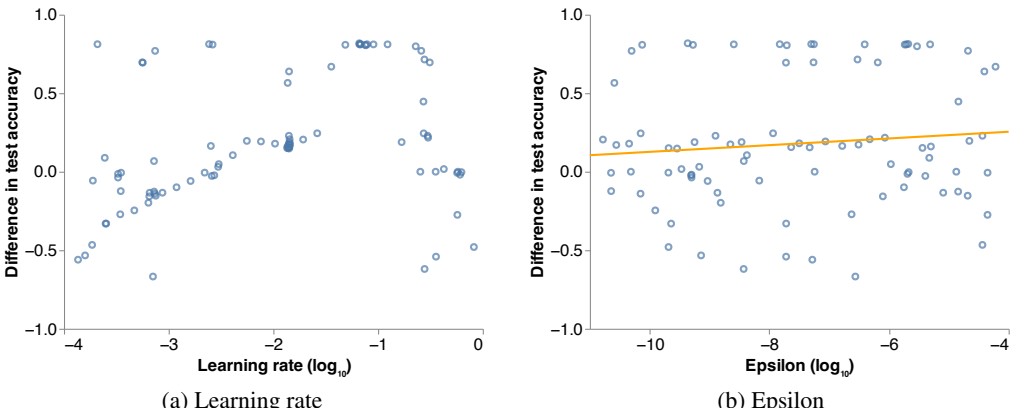

(a) Learning rate

(b) Epsilon

Figure S2: Raw results for case study 1. Difference in test accuracy (SGD - Adam) by (a) learning rate and (b) $\epsilon$. Orange line fit with linear regression for illustration purposes

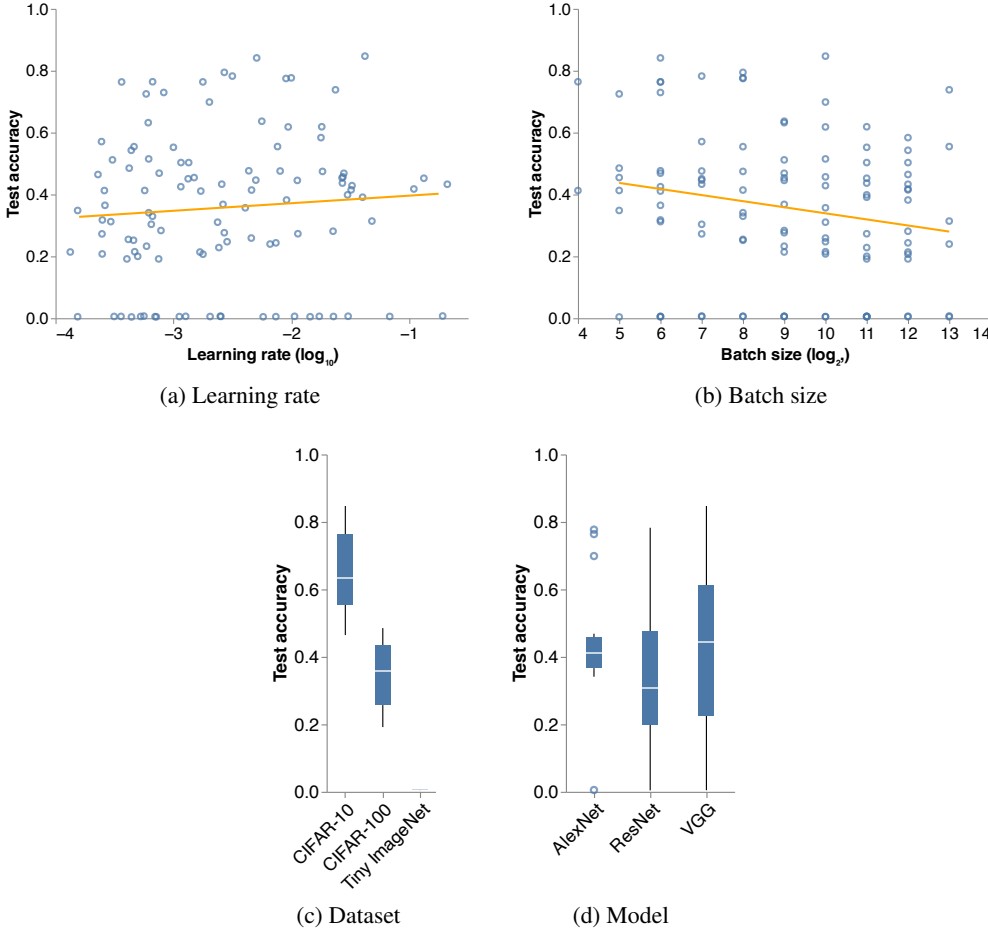

(a) Learning rate

(b) Batch size

(c) Dataset

(d) Model

Figure S3: Raw results for case study 2. Test accuracy by (a) learning rate; (b) batch size; (c) dataset; (d) model. Orange lines fit with linear regression for illustration purposes

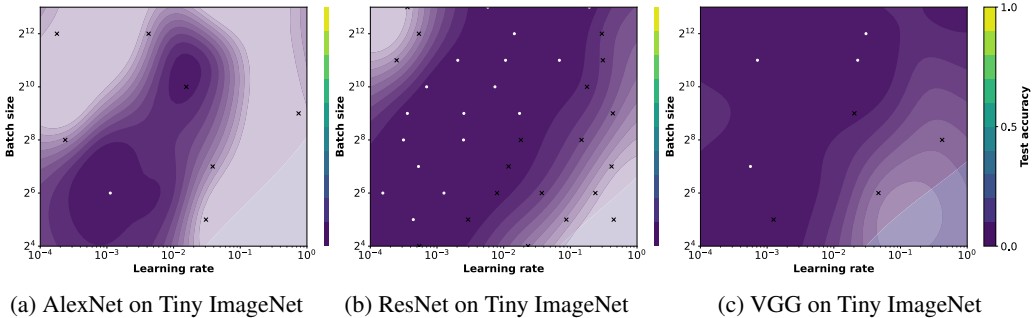

(a) AlexNet on Tiny ImageNet    (b) ResNet on Tiny ImageNet    (c) VGG on Tiny ImageNet

Figure S4: Contour plot of GP-predicted mean test accuracy over the search space of learning rate, batch size, and model for Tiny ImageNet only. White points are trials with training accuracy $\geq 0.99$; black crosses were excluded.

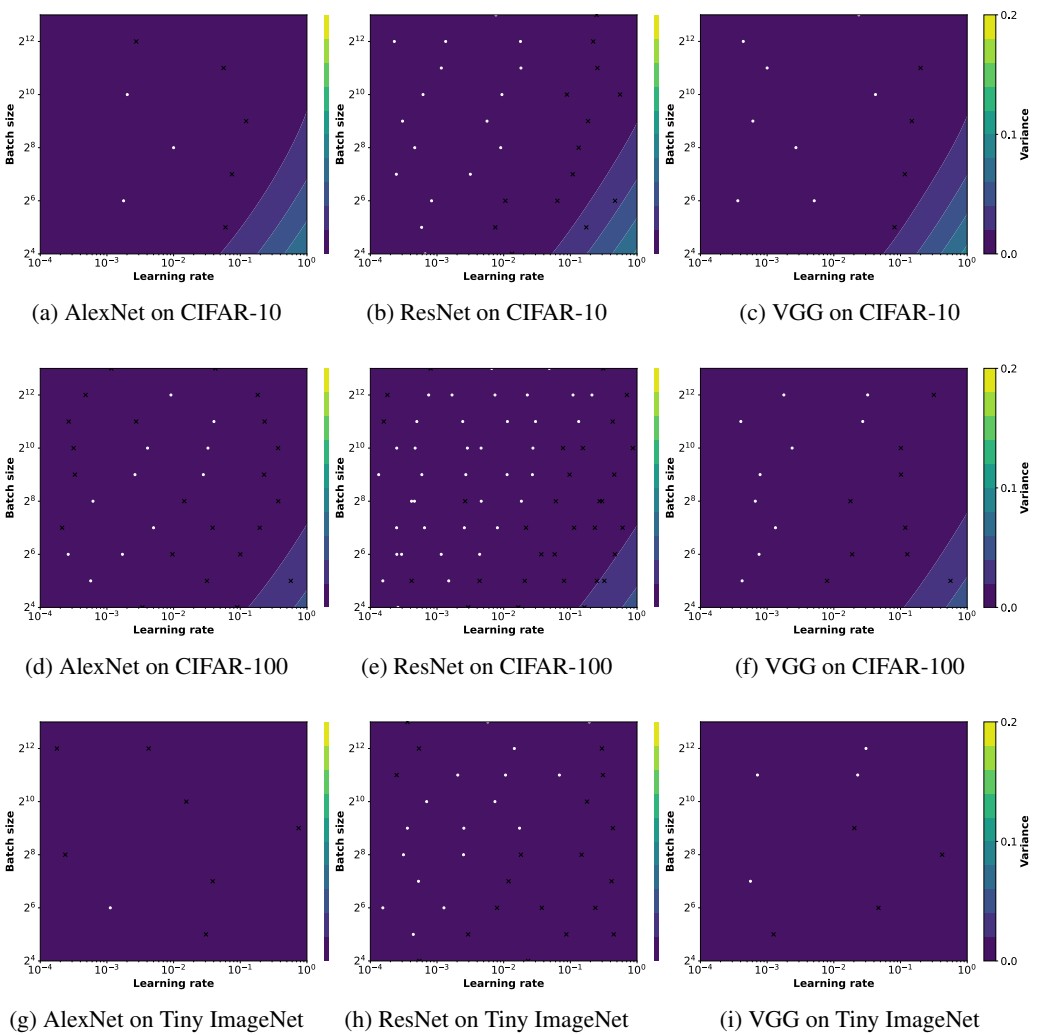

Figure S5: Contour plot of test accuracy variance over the search space of learning rate, batch size, dataset and model. White points are trials with training accuracy $\geq 0.99$; black crosses were excluded.

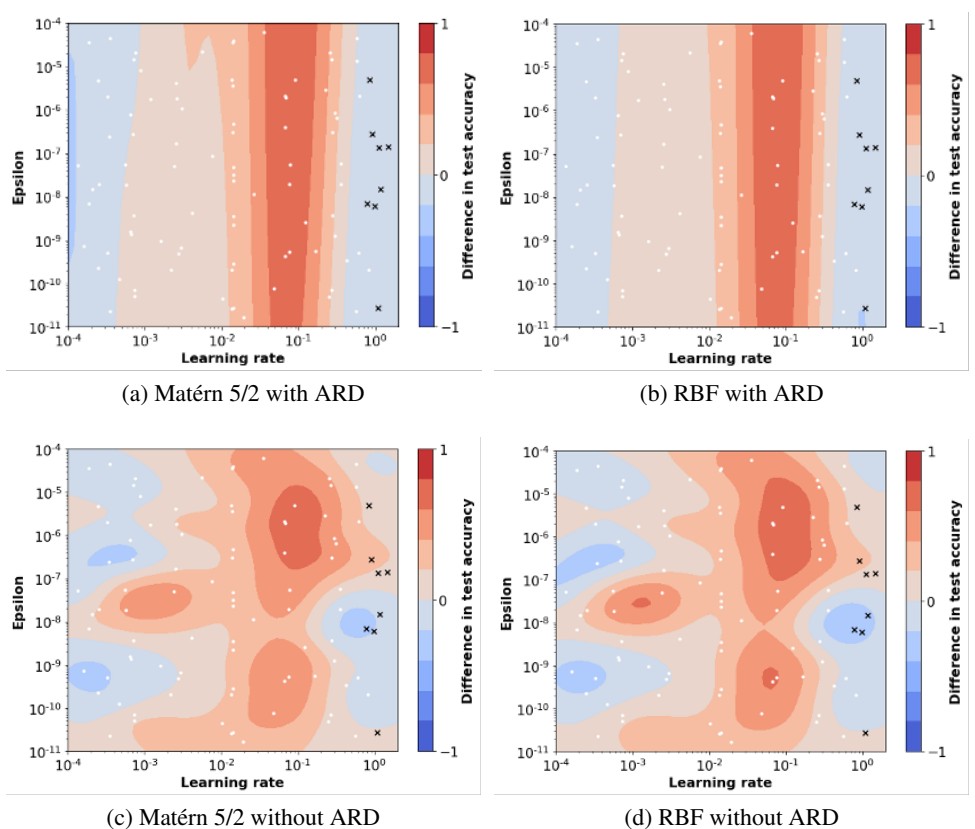

(a) Matérn 5/2 with ARD          (b) RBF with ARD

(c) Matérn 5/2 without ARD          (d) RBF without ARD

Figure S6: Effect of GP kernel on predicted mean difference in test accuracy (SGD - Adam) over the search space of learning rate and $\epsilon$. Different kernel choices may impact the contour visualization, but do not change the primary conclusion regarding learning rate sensitivity and $\epsilon$ insensitivity. A complementary analysis over different initial kernel lengthscales and variances yielded Bayes factors consistent with our conclusions across all trials, showing that the conclusions we draw are insensitive to GP hyperparameters.

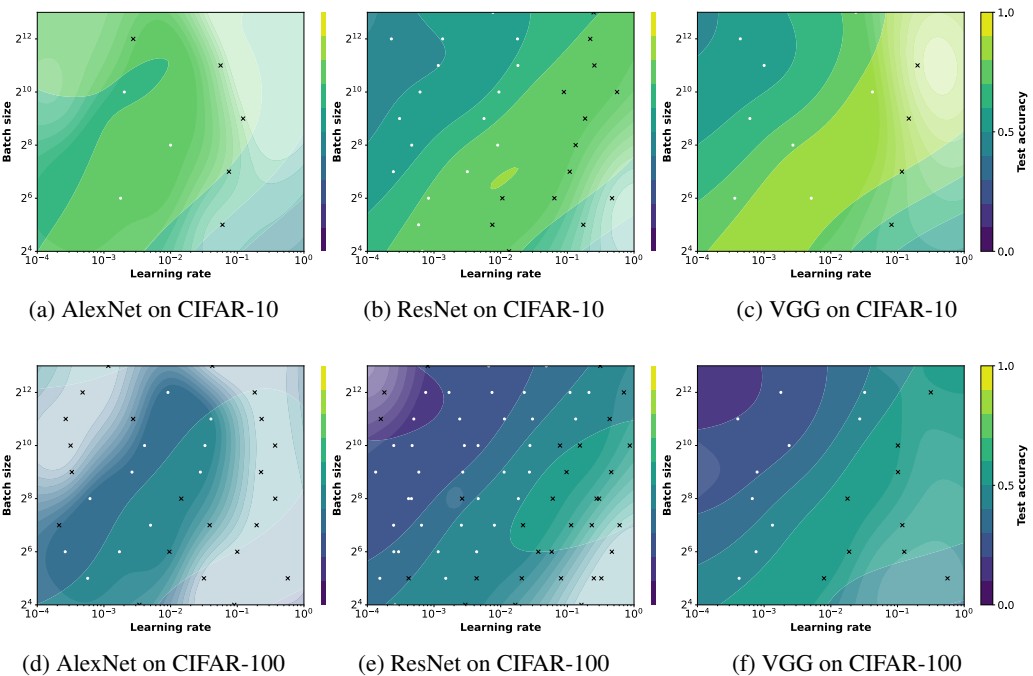

Figure S7: Effect of GP kernel on predicted test accuracy over the search space of batch size, learning rate, model and dataset: **Matérn 5/2 kernel with ARD**. Contours are broadly consistent across kernel choices as shown in figs. S8 to S10. This figure is repeated in the main text, but included here for easy comparison.

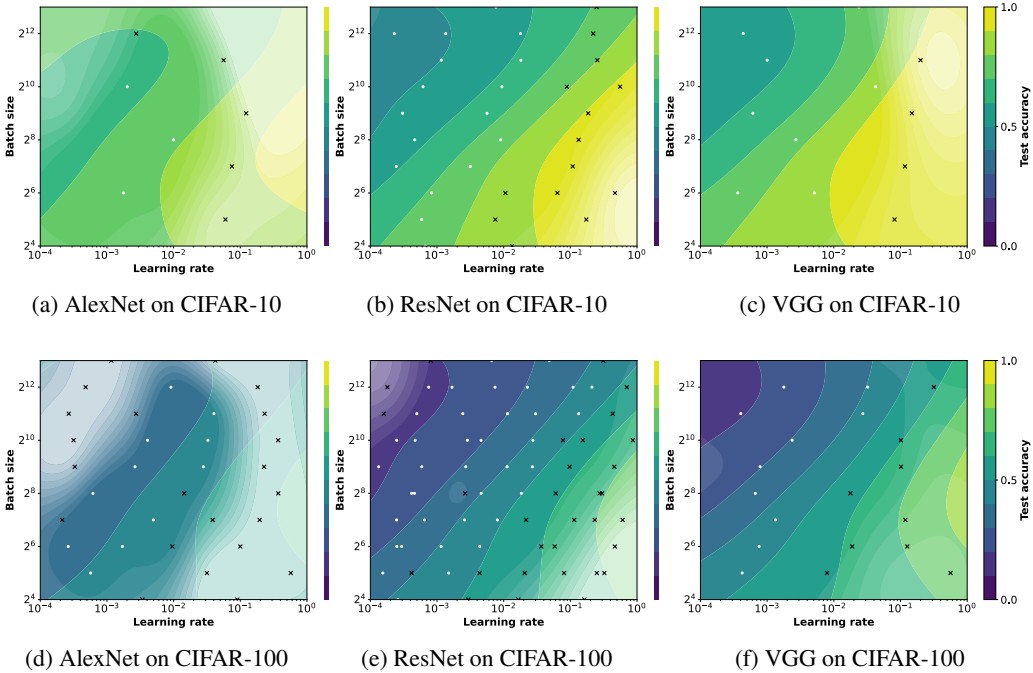

Figure S8: Effect of GP kernel on predicted test accuracy over the search space of batch size, learning rate, model and dataset: **Matérn 5/2 kernel without ARD**.

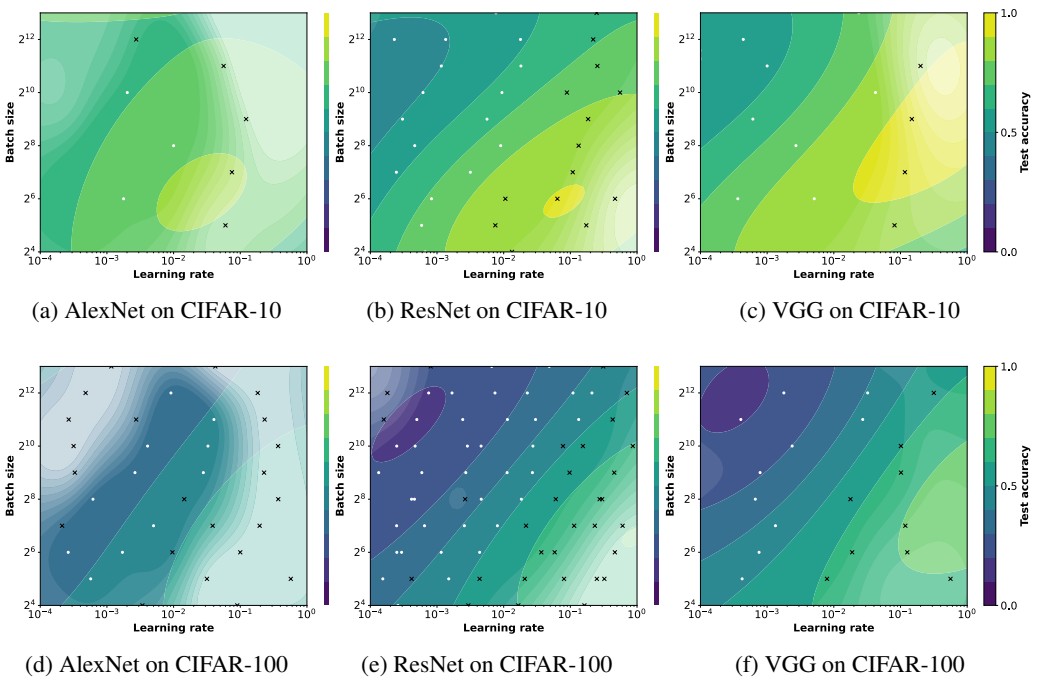

(a) AlexNet on CIFAR-10  (b) ResNet on CIFAR-10  (c) VGG on CIFAR-10

(d) AlexNet on CIFAR-100  (e) ResNet on CIFAR-100  (f) VGG on CIFAR-100

Figure S9: Effect of GP kernel on predicted test accuracy over the search space of batch size, learning rate, model and dataset: **RBF kernel with ARD**.

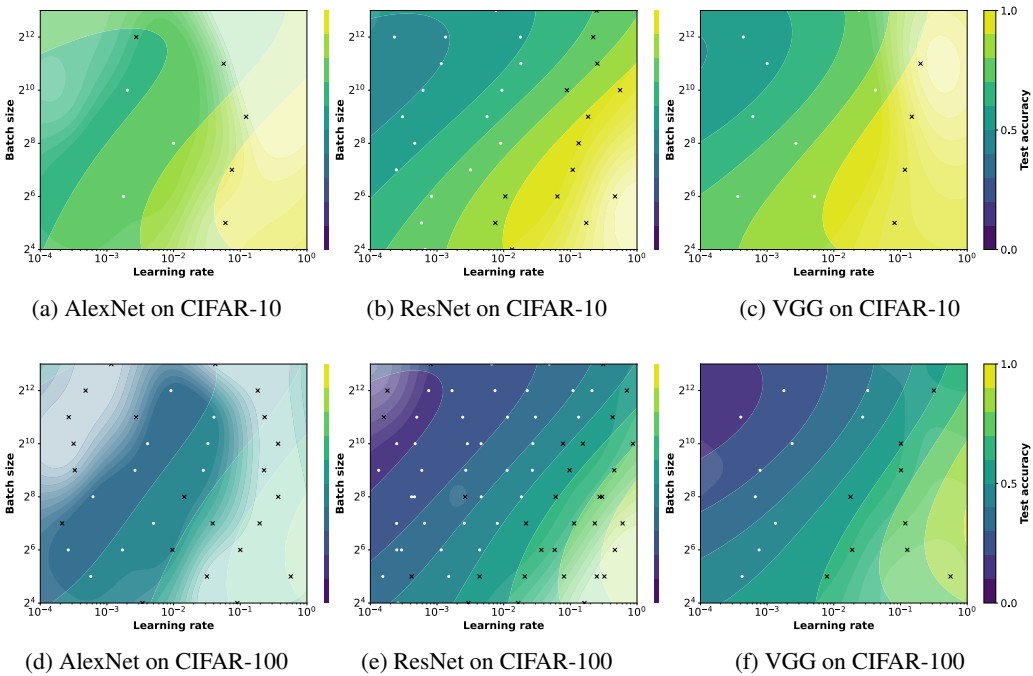

(a) AlexNet on CIFAR-10  (b) ResNet on CIFAR-10  (c) VGG on CIFAR-10

(d) AlexNet on CIFAR-100  (e) ResNet on CIFAR-100  (f) VGG on CIFAR-100

Figure S10: Effect of GP kernel on predicted test accuracy over the search space of batch size, learning rate, model and dataset: **RBF kernel without ARD**.