# OpenReview forum: "Modeling the Machine Learning Multiverse"
_NeurIPS.cc/2022/Conference — NeurIPS 2022 Accept_

### Official Review · Reviewer_yyG7 · 2022-07-11

**Rating:** 7
**Confidence:** 4
**Soundness:** 3 good
**Presentation:** 3 good
**Contribution:** 3 good

**Summary:**

This paper proposes an ML multiverse analysis framework to draw robust scientific conclusions about model performance by focusing on the model exploration using Gaussian process surrogate models and Bayesian experimental design. The paper shows two motivating case studies:
1) presented a multiverse analysis demonstrating that different hyperparameter choices lead to different optimizer recommendations;
2) used the framework to investigate the conflicting research on the effect of learning rate on the large batch training generalization gap.

The core idea of the ML multiverse analysis is to use integrated variance reduction (IVR) as the acquisition function and keep all available information to draw more bust conclusions. The analysis is carried out by visualizing the posterior predictive mean to the extent to which the surrogate's posterior variance has been adequately explored. Also, the analysis tests for the interaction effects of the different dimensions of the search space by calculating the Bayes factor of a GP with shared kernel against an additive kernel. Finally, a Monte Carlo sensitivity analysis is used to test how much a change in one of the parameters would influence the model outputs.

**Questions:**

-  As pointed out by the authors, the largest dimension of the multiverse in the two case studies is 4, therefore, faceted contour plots are sufficient for visualization purpose. In case of high-dimensional search space, besides visualization, are there any other quantitative techniques can be used to analyze the result?

**Limitations:**

The author did addressed the potential environmental concern of the framework due to computation scaling up with the search space. On the other, scalability of the proposed method with respect to the search space will probably limit the practicality of the proposed multiverse analysis framework.

**Strengths And Weaknesses:**

## Strength:
- The paper is well written and structured
- The paper provides a principled and practical way to to achieve reproducible results and draw robust conclusions, which will help the continued progress in the ML community.
- The effectiveness of the proposed framework is well-demonstrated by two comprehensive case studies.

## Weakness:
- As pointed out by the author, the computation time of the framework will scale with the multiverse size, given the recent critiques of environmental impacts on ML, needs to be carefully considered. One possible way to reduce the computation time is to make use of all the existing experiments results through efficient surrogate models.

---

> ### Author Response · Authors · 2022-08-02
> **Response to reviewer yyG7 on compute cost and analyzing multiverse results**
>
> Thanks once again for the considered and constructive review. We have included two comments below to respond to your comment about compute cost and your question about high-dimensional analysis.
>
> ## 1. On compute cost and environmental impact:
>
> We agree that compute cost and environmental impact needs to be carefully considered in all contemporary machine learning work. However, as we note on lines 284-5, we posit that our approach need not use substantially more compute that existing hyperparameter optimization schemes: instead, by switching from an optimization- to an exploration-based approach, researchers can spend a roughly equivalent amount of compute exploring in place of their traditional hyperparameter search.
>
> Moreover, on aggregate over a whole community, we argue that the added transparency around results and improved robustness of conclusions can indeed reduce net cost, by supporting researchers to spend their budget wisely on new experiments, rather than repeating past experiments performed by different researchers whose results may not have been reported.
>
> Finally, we are not advocating for the blanket adoption of the multiverse analysis in all scenarios. There may be scenarios where added compute may deliver little value, though in others — e.g. high-stakes settings such as clinical applications — the added benefit of more trustworthy and robust conclusions will likely outweigh the cost.
>
> ## 2. On summarizing and visualizing multiverse results:
>
> We think novel strategies to summarize main conclusions in a multiverse setting are an exciting area for future research. As we mention in lines (299-301), we see promise in interactive multiverse visualization, allowing the reader to slice up the multiverse into chunks most relevant for their interests.
>
> Moreover, in higher-dimensional settings, standard dimensionality reduction techniques might be able to highlight emergent trends. A concurrent and related work in neuroimaging [1] (its pre-print form was also mentioned by reviewer wcgM) applies multi-dimensional scaling (MDS) to visualize a high-dimensional search space in 2D.
>
> We also highlight our application of Monte Carlo sensitivity analysis to the GP parameters which demonstrates how robust conclusions are in the face of variation along each dimension. Reporting the results of this sensitivity analysis can scale to larger and more complex search spaces. Furthermore, we expect that in the majority of cases, only a subset of dimensions will be identified as relevant by the sensitivity analysis, enabling more detailed visualization and analysis of just the dimensions that matter.
>
> Finally, we suggest that a key idea in the multiverse analysis as presented here is to move beyond summarization and towards increased, even radical, transparency. In our community, we often focus on a single conclusion or a top-line figure. Instead, we suggest it is important to transparently report results over the full (or at least, larger) space, such that future researchers and practitioners can navigate through and understand the different subspaces in which results hold and in which they differ.
>
> [1] Dafflon et al. A guided multiverse study of neuroimaging analyses. Nat Commun. 2022 Jun 29;13(1):3758.

---

### Official Review · Reviewer_dGQ4 · 2022-07-11

**Rating:** 6
**Confidence:** 3
**Soundness:** 3 good
**Presentation:** 3 good
**Contribution:** 3 good

**Summary:**

The paper adopts a concept of psychology, called ‘Multiverse Analysis’, to systematically study credibility of claims and robustness of conclusions. Multiverse are universes each with a slightly different layout. Based on this concept, GP model is employed to quantify and emulate the relationship between interest of study and variance in layouts. The whole experiment is conduct in a Bayesian way: construct initial design; collect data; find next experiment point that reduce the variance. The existence of interaction effect is determined by comparison of probability of the model considering interaction effect and the model not given current data.

**Questions:**

How to verify the significance of model comparison (interaction effects)? If the Bayes factor is close to 1, say 0.998.

**Ethics Review Area:**

["I don’t know"]

**Limitations:**

The authors have discussed the limitations. I have one more comment.

For the IVR acquisition function in Eq (6), the authors just simply use a Monte Carlo approximation to deal with multivariate integral. It cannot be accurate when for large-dimension problem.

**Strengths And Weaknesses:**

Strength:
(1) It is very novel to consider investigating the machine learning multiverse as the design of experiments problem with the steps of initial design, surrogate model, acquisition function, and sequential design.

(2) The proposed framework is capable of comprehensively verify or study an interested claim of effect. Bayes factor is a fantastic idea to determine if interaction effect exists.

Weakness:
(1). The current method in the manuscript is not able to quantify the magnitude of interaction effects. When model the categorical factor, kernel function is a production. Therefore, the effects are kind of interaction effect among all categorical factors and continuous factors.

(2). The current manuscript has not sufficiently described how to choose the initial design, and how the choice of the initial design affects the performance of the proposed method.

---

> ### Author Response · Authors · 2022-08-02
> **Response to reviewer dGQ4 on interaction effects, initial design and search space choice, MC integration and Bayes factor**
>
> Thanks for the thought-provoking questions. We’ve responded to your questions below.
>
> ## 1. On quantifying interaction effects & categorical parameters:
>
> The use of Bayes factor does test for the conclusive existence of an interaction effect, though the reviewer is correct that it will not establish the magnitude of the interaction. Instead, we turn to Monte Carlo sensitivity analysis to estimate the relative importance of each dimension, and of their interaction by way of the total effect (e.g. fig. 3b and 6b).he
>
> We also wish to clarify our choice of model when using categorical parameters. We explicitly choose an intrinsic coregionalization model (lines 197-200), rather than the standard product kernel for categorical parameters. This models each combination of categorical parameter value as its own output function, which enables two things. First, we are able to evaluate the precise effect of categorical parameters, e.g. in fig. 5a where we show that dataset choice can influence conclusions. Second, it allows us to identify shared variance between the different outputs. In this way, our interaction effect test (i.e. comparing additive/shared _base_ kernel models) tests for interaction effects across all model/dataset pairs.
>
> ## 2. On initial design:
>
> There are two “initial designs” the reviewer might be referring to: how we draw the initial set of points, or how we determine which dimensions to include in the analysis and how to bound them.
>
> ### 2.a. On sampling the initial points:
>
> For our initial set of points, the best set of points will be one that optimally covers the space, i.e. one with low discrepancy. For this purpose, the Sobol sequence is close to optimal, and there are existing implementations that are available out-of-the-box in contemporary bayes opt / experimental design libraries.
>
> That said, even sampling uniformly at random for the initial points is a reasonable strategy. The main motivation behind low discrepancy sequences is to avoid undersampling from large regions of the search space, but this is (a) relatively unlikely and (b) low-impact as any gaps would be immediately filled in by the points selected by IVR.
>
> We are primarily concerned with whether conclusions about model performance are reproducible, and we believe sampling uniformly at random vs. using the Sobol sequence will not change the outcome here.
>
> ### 2.b. On defining a search space:
>
> The search space necessarily impacts conclusions drawn. Just as with any experiment, effort and consideration should go into the search space design - indeed, this is the trickiest part of the multiverse analysis.
>
> Candidate dimensions for inclusion are all those that the researcher suspects are “relevant”. In other words, the dimensions should explicitly codify auxiliary hypotheses about the conclusion(s) being drawn: assumptions about when results should hold, and when we expect them to break down.
>
> There isn’t an easy answer as to how to choose a space, and it requires judgment on the part of the researcher, and specialist knowledge about the domain under consideration. In short, defining a search space is indeed hard, but most attempts (if accompanied by the requisite transparency) are far better than nothing.
>
> ## 3. On MC integration
>
> In our work, we use MC to approximate the integral over the search space. This is the standard approach to integrated variance reduction and is commonly used. As we’re only dealing with a handful of dimensions in our case studies, we think this is a sufficient approach to demonstrate the utility of an exploration-only acquisition function like IVR. In larger settings, an alternative sampling approach such as a Sobol sequence or other quasi-MC methods might be more appropriate, and this is indeed a design choice (though perhaps only for really large multiverses). We’ll definitely add something to this effect to the discussion section..
>
> ## 4. On Bayes factor model comparison
>
> In this specific case, we use Bayes factor to test for interaction effects by comparing the model fit between a GP with an additive kernel and a GP with a product kernel. We note that this is specific to the question at hand, i.e. whether there exists an interaction effect between e.g. LR and epsilon, and is not a general property of all multiverse analyses: future studies may choose other tests that are appropriate to the claims at hand.
>
> In interpreting the Bayes factor, a result close to 1 would indicate an equivocal result, i.e. that it is not conclusive whether there is or is not an interaction effect. In this case, a possible response would be to collect more data until a conclusive result is obtained. Indeed the applicability of optional stopping is a key motivation for using a Bayesian statistical hypothesis test as opposed to a frequentist. Collecting data until a hypothesis is (dis)confirmed up to a satisfactory degree of confidence is a useful secondary outcome of the multiverse framework.

---

### Official Review · Reviewer_fyXL · 2022-07-12

**Rating:** 6
**Confidence:** 4
**Soundness:** 3 good
**Presentation:** 3 good
**Contribution:** 3 good

**Summary:**

The paper proposes the Multiverse analysis as a step toward more transparent, reliable and credible results when we are facing large number of choices in our research design even before running the actual experiments. Speficically the authors leverage Bayesian optimization idea to develop a framework to search over large search space, which is used to guide honest research designs.

**Questions:**

1. As the authors stated in the paper, someone would question the fact that search space is also a choice. I agree with the explanations given by the author, however, I think the choice of kernel (beyond the two in the paper) and the prior (e.g. $\Sigma$) would also make a difference, especially when the budget is limited as we cannot run many steps. Do you have results showing robustness of your conclusion to these choices.
2. I am not sure what the "appropriate stopping criterion" in step 4 is. Is it just a visualization as stated in line 82?

**Limitations:**

The authors addressed their limitations and negative societal impact.

**Strengths And Weaknesses:**

Strengths: The problem itself is really important and the authors put a lot of effort in reproducibility, providing the code for running the experiments. Also, the presentation is very clear and the three examples chosen clearly convey the message the authors want to share.

Weaknesses: My main concern is whether this is just a straightforward application of Bayesian Optimization in parameter tuning when the parameters are replaced by different research designs in this setup. Some other concerns are listed in the questions.

---

> ### Author Response · Authors · 2022-08-02
> **Response to reviewer fyXL on differences from BO, GP hyperparameter robustness and appropriate stopping criterion**
>
> Thanks for the helpful review. We’ve uploaded a new version of the paper with an additional figure in the appendix addressing your question about GP hyperparameters, and responded to your questions below.
>
> ## 1. On differences from Bayesian Optimization
>
> The primary contribution of our paper is to introduce the multiverse analysis to the machine learning community, in order to support drawing robust and generalizable scientific conclusions. To make the multiverse analysis tractable, we couple this approach with a GP surrogate and bayesian experimental design.
>
> While our approach builds on the wealth of literature in Bayesian Optimization, the goal of our approach is in fact not optimization, but rather exploration. This is a fairly fundamental difference, and we view it as a benefit that our approach has this connection, as future work building on the multiverse analysis can leverage the extensive machinery developed for BO. As the field of BO progresses, the multiverse analysis directly benefits.
>
> To reiterate, a key takeaway from our paper is that while there is of course a role for optimization in machine learning, the role for exploration is often underestimated. Indeed, for those conducting research into new models, or exploring how and when models work, exploring the multiverse of choices systematically can provide much needed transparency that is of significant value to downstream researchers and practitioners.
>
> Finally, we note that our framework extends far beyond hyperparameters. Indeed, in case study 2 we explore (although at a small scale) the effect of both dataset and model choice on the generalization gap phenomenon. In lines 274-276, we suggest that an expanded search space could even include the evaluation metric and the termination criterion. While _optimizing_ over datasets and evaluation metrics would certainly be bad scientific practice, exploring them is exactly the converse: demonstrating robustness in the face of these choices would greatly reinforce the contributions of our community. Our case studies are just that; they are examples designed to illustrate the potential of the multiverse analysis as principled framework: the most exciting part of our research is not in what we have done so far, but in the possibilities that lie ahead.
>
> ## 2. On robustness to GP hyperparameters
>
> We think this is an excellent question: just like with search space, we fully appreciate that choosing how to do a multiverse analysis at a technical level is of course a choice.
>
> However, we have intentionally selected the Matern 5/2 kernel with ARD because it should be a sensible out-of-the-box choice for most use cases, and indeed is often used in Bayesian Optimization (which faces exactly this question also) for this reason.
>
> We have also ran a few analyses on case study 1 and added a figure to the appendix (fig. S6) showing the minimal impact of the kernel on the contour plot: the plots change a little, but the broad conclusions remain consistent. We also analyzed a range of initial variance and lengthscale hyperparameters and find no change in the the Bayes factor outcome for case study 1. We think this is useful evidence supporting the general applicability of these choices. We will elaborate upon this idea and extend it to case study 2 for the camera-ready if accepted.
>
> ## 3. On appropriate stopping criterion
>
> Thanks for the great question. We describe the stopping criterion on lines  87-89: we continue sampling from the multiverse until we reach a conclusive Bayes factor. We acknowledge that “appropriate stopping criterion” is a little hand-wavy. We use this terminology because stopping criterion is application specific - for other researchers in different contexts, it could be a fixed number of experiments, or a certain amount of compute. We will endeavor to tighten up this idea in the final version.

---

> > ### Comment · Reviewer_fyXL · 2022-08-07
> > **Response to Rebuttal**
> >
> > Thanks for the response. I think my concerns are well addressed.

---

### Official Review · Reviewer_wcgM · 2022-07-12

**Rating:** 7
**Confidence:** 4
**Soundness:** 4 excellent
**Presentation:** 4 excellent
**Contribution:** 2 fair

**Summary:**

The paper proposes to use active learning with Gaussian Processes (GPs) as a way to perform sample-efficient "multiverse analysis" of machine learning models, i.e. a way to understand the full space of hyperparameter settings and their influence on model performance rather than simply optimizing them. It then applies this analysis to three examples, one with a motivating (but artificial) goal related to inter-group variability, and two motivated by real recent debates in the literature.

**Questions:**

I hope that the rebuttal can help clarify and justify the impact of the findings as presented, for example by more precisely discussing ways in which they drive new conclusions not present in the cited prior work. Alternatively, to the extent that the conclusions are similar in broad strokes, I hope the rebuttal can defend the ways in which the systematicity of the present experiments shores up some aspect of the two case studies that remained previously unclear or murky. A third way for the paper to have greater impact would be to demonstrate that additional conclusions can be drawn from the meta-models that extend beyond what has been hypothesized or shown previously.

Finally, I also hope that the rebuttal will clarify the following puzzle regarding the adaptive optimizer study: on lines 167-170, the paper discusses how Adam's epsilon parameter seems to not matter for the purposes of determining whether it outperforms SGD, and grants that epsilon could be tuned to improve Adam's performance. If that is true, then one can take a vertical slice through figure 2a and move along it in some dimensions in which Adam performance improves but SGD performance does not (since SGD performance should be invariant to movement in that dimension of the space). To the extent that we do not see such a thing, the present contribution seems incompatible with the claim that epsilon can be used to improve Adam's performance, or be used to close the Adam-SGD gap. What am I missing here?


**Limitations:**

Limitations are discussed, even alongside the claims / conclusions rather than being shunted to a separate section -- this is good. Benefits of using sample-efficient active learning are discussed. Negative societal implications are not discussed explicitly, though I don't think there's anything major to discuss here. One thing not mentioned that is possibly worth engaging with is that for very large-scale models that take thousands of hours to train on thousands of GPUs, even regular hyperparameter search may be out of reach, never mind a multiverse analysis, so the impact of this sort of work will likely be in smaller settings potentially farther from state of the art.

**Strengths And Weaknesses:**

## Strengths
I enjoyed reading the paper: it is well-written such that I have essentially no quibbles related to typos, confusing wording, or the like, and more importantly the motivation and core ideas are presented so clearly as to seem obvious post hoc. The idea of using a meta-model to improve efficiency of multiverse analysis is likewise useful (though I believe there's a similar idea in a biorxiv preprint from a few years ago, doi:10.1101/2020.10.29.359778, and in older work from psychology doi:10.1037/0033-295X.113.1.57). The technical approach is reasonable given the setup (flexible GP surrogates, global active learning with IVR by taking an expectation over the lookahead posterior variance), and the conclusions aren't overstated given the relatively modest studies undertaken.

## Weaknesses
The essence of a good paper is an insightful idea or contribution taken to new payoffs, and both seem relatively modest in the paper as it stands.

With regards to the core idea, the paper imports a pre-existing conceptual framework (in the sense that multiverse analysis is not new, though it seems new to ML), and uses standard tools in only a slightly novel application (in the sense that GP active learning is not new in ML, though the typical application is to pure optimization).

With respect to the new payoffs, the analysis of adaptive optimizers comes to a similar high-level conclusion to what is in prior work (that both SGD and Adam can perform equally well depending on hyperparameter tuning), and the paper does not go beyond exploring the two parameters that have been explored previously. At the same time, when it comes to the details, the paper does not seem to replicate the effect of epsilon on the SGD-Adam gap. This is potentially interesting and while the paper claims this result is not incompatible with the potential to improve Adam performance by tuning epsilon, I'm not sure I see how both can be compatible (more on this in "questions" below).

With the large batch generalization gap study, the paper explores a space substantially smaller than the multiverse of reasonable choices previously explored in the literature, and the relationship it does find matches prior work regarding batch size and learning rate,

Without disputing that the present work undertakes both studies in an elegant and efficient way relative to prior work, I think the latter still serves as a "warmup" demonstration of shoring up existing results, and not a strong new thing that couldn't have been done without the proposed approach, while the former adds another datapoint to the space without providing a definitive understanding of the benefits of Adam over SGD.

Furthermore, the small search spaces and large batches that the paper considers don't really showcase the benefit of active learning where, especially for a pure exploration setting, a quasi-random search baseline is likely to be quite strong.

---

> ### Author Response · Authors · 2022-08-02
> **Response to reviewer wcgM on contribution, large-scale training, case study choice and Adam's epsilon**
>
> Thanks for the useful comments. Our specific responses are below.
>
> ## 1. On contribution and novelty
>
> We see our core contribution as introducing the multiverse analysis as a principled framework to the machine learning community, where it can add significant and long-term value by promoting the reporting of robust and generalizable conclusions.
>
> While we did not invent the multiverse, we believe that importing ideas from other disciplines can be of immense value to the machine learning community. In this specific case, we hope to demonstrate concretely how the multiverse analysis is relevant to machine learning and to make it practical with a GP surrogate and Bayesian experimental design.
>
> To the reviewer’s point about active learning, we are of course the beneficiaries of the long heritage of GP-enabled approaches across bayesian optimization, experimental design and active learning. One novel contribution here is to marry the multiverse up with a GP + Bayesian experimental design to enable efficient exploration. The mentioned neuroimaging pre-print [1] has only just (June 29th) been published in Nature Comms, and as such we consider this a concurrent work in an entirely separate field, though we will of course update our related work for the final version if accepted.
>
> ## 2. On very large-scale models
>
> We partially agree with this critique, and acknowledge that this kind of systematic exploration isn’t suitable for all research.
> Beyond model training: we draw attention to the vast array of empirical evaluations on various topics throughout machine learning research. The multiverse as principled framework is equally applicable when one is evaluating a pre-trained, off-the-shelf model and its performance(s) on various datasets, under different conditions.
>
> Specifically regarding model training: precisely because training very large-scale models is expensive, ML practitioners typically spend a lot of time and compute doing a hyperparameter search before training the full model. For example, a recent paper "What Language Model to Train if You Have One Million GPU Hours?" (https://openreview.net/pdf?id=rI7BL3fHIZq) is a great example of a hyperparameter search that would have benefitted from a multiverse analysis.
>
> If, on the other hand, hyperparameter search is infeasible, then we would argue that this is a (possibly transient) limitation of massive scaling, rather than of the multiverse analysis. If models are to require significantly more investment in compute and carbon, and are to be deployed in increasingly important scenarios, we think it’s all the more valuable to analyze their robustness in a principled way.
>
> ## 3. On case study scale
>
> Our two principle case studies are designed to illustrate, separately, a “reproducibility” example and a “synthesis” example.
> For the former, case study 1, we chose our search space — admittedly small — to demonstrate the value of analyzing a space rather than a point estimate: it is not the aim of this paper to conclusively adjudicate between optimizers, but to highlight that the impact of learning rate changes the conclusions drawn by Wilson et al. [2]: the “best” optimizer varies by learning rate.
> For the latter, case study 2, we chose a slightly larger search space to demonstrate how our analysis can generalize to non-hyperparameter settings such as dataset and model. A goal of this case study is to show that the multiverse analysis is not just for reproducibility, but can also help synthesize a large and complex topic across many papers and even more experimental settings (although, as we note in the paper, a fuller analysis would be welcome).
> These examples are exactly that: case studies designed explicitly to showcase the multiverse and to demonstrate our general framework, and we don’t feel it is justified to spend more on compute for the sake of examples that would only differ in degree rather than in kind. We are, however, tremendously excited about the potential for subsequent work to apply the multiverse to larger domains and more complex questions, particularly in domains where robust conclusions have real–world impacts, such as medical applications or scientific discovery.
>
> ## 4. On Adam’s epsilon
>
> In lines 167-170, we are just acknowledging the possibility that when *optimizing* over both LR and epsilon, one could potentially improve Adam’s performance: we don’t at any point dispute that Adam’s performance can’t be improved by tuning epsilon. Instead, the principle claims investigated with the multiverse analysis are with respect to the relative performance of Adam and SGD, and we suggest that *LR is the driving factor* in determining relative merit.
>
> [1] Dafflon et al. A guided multiverse study of neuroimaging analyses. Nat Commun. 2022 . 13(1):3758.
>
> [2] Wilson et al. The marginal value of adaptive gradient methods in machine learning. NeurIPS, 2017.

---

> > ### Comment · Reviewer_wcgM · 2022-08-06
> > **Response to rebuttal**
> >
> > I buy the arguments regarding very large scale models and novelty. With respect to the case studies chosen, while it's true that the scale doesn't showcase the benefits of GP active learning, I agree that that's not important to the paper. What is important is that they don't show something new we didn't have evidence for before, which would make for a stronger paper -- I don't think the rebuttal disputes this. With respect to the Adam epsilon parameter, I agree that the paper does not make an explicit claim about performance, but I am still puzzled on how to interpret the figure in light of the claims in the text (see my original review). I hope the authors and / or the other reviewers will help me puzzle this out.
> >
> > Regardless, I think neither issue sinks the paper, and I will update my rating accordingly.

---

> > > ### Author Response · Authors · 2022-08-09
> > > **Response regarding question about epsilon**
> > >
> > > Thanks for your response. If figure 2.a. is still causing confusion, that’s helpful feedback that it could be clearer. We’ll do our best to tighten this up going forward.
> > >
> > > For now, just to check we’re on the same page, we interpret your question about epsilon as follows: How could epsilon be used to improve Adam’s performance, given that fig 2.a. shows that one can move up and down the y axis (epsilon) almost freely with barely any change in relative performance?
> > >
> > > We hope this is a fair and accurate representation of your question. If so, then our response is as follows:
> > >
> > > The relative performance contour plot in fig. 2.a. is only an estimate of the true space. Our intent with the comment above is to say that if one continued to sample _at length_, and were to sample more densely around a specific region (i.e. via hyperparameter optimization), then perhaps the resulting _more detailed_ view of the LR x epsilon search space might reveal some _very modest_ room for improvement by tuning epsilon.
> > >
> > > Crucially, this is not in conflict with our claim that LR determines the relative merit of Adam/SGD, and that any effect of epsilon is dwarfed by that of learning rate. Indeed the sensitivity analysis in fig. 3. estimates a minuscule effect of epsilon. We are not suggesting that epsilon can “be used to close the Adam-SGD gap”. Instead, we are simply not _ruling out_ that tuning epsilon might eke out a (tiny) drop more performance when using Adam.

---

> > > > ### Comment · Reviewer_wcgM · 2022-08-10
> > > > **Understood**
> > > >
> > > > Makes sense, though if your interpretation is correct, should we expect the reported gap between SGD and Adam to fit within the range of the surrogate model's posterior uncertainty? Not critical, but would be another sanity check on that claim.

---

### Author Response · Authors · 2022-08-02
**Thanks for your detailed and positive reviews**

Thank you to all four anonymous reviewers for your helpful and constructive comments.

We loved hearing that you found our work very novel (dGQ4), principled and practical (yyG7), are satisfied with our technical approach (wcgM) and particularly that you found using the Bayes factor to test for interaction effects a “fantastic idea” (dGQ4). We sincerely appreciate your kind feedback.

We are also pleased to see that you enjoyed reading (wcgM) the paper, that you found it well-written (wcgM, yyG7), and particularly that you found the core ideas well-presented (wcgM, fyXL) and well-demonstrated by our case studies (yyG7). We—of course—wholeheartedly agree that the problem is important (fyXL); indeed our core motivation is to facilitate more solid foundations for “continued progress in the ML community” (yyG7).

We’ve added in-depth responses to each of your reviews below, and hope they serve as useful clarification and highlight how we think about the multiverse. We look forward to continuing the discussion with each of you.

---

### Meta-Review · Area_Chair_3LL7 · 2022-08-23

**Recommendation:** Accept
**Confidence:** Certain

**Metareview:**

This paper suggests a novel way to conduct machine learning empirical research and report the results. While ablation studies became a common practice in analyzing the contributions of different components in the ML system, this paper takes it much further by suggesting a way to explore the entire hyper-parameter space.
While the components used in the work (Gaussian Processes, Active Learning, …) are not new, the novelty of this work is in combining them into addressing a timely question. This paper has the potential to contribute to the way ML research is conducted and reported.


**Award:**

No

---

### Decision · Program_Chairs · 2022-09-14

Accept